# In Vivo Evaluation of Two Hemorrhagic Shock Resuscitation Controllers with Non-Invasive, Intermittent Sensors

**DOI:** 10.3390/bioengineering11121296

**Published:** 2024-12-20

**Authors:** Tina M. Rodgers, David Berard, Jose M. Gonzalez, Saul J. Vega, Rachel Gathright, Carlos Bedolla, Evan Ross, Eric J. Snider

**Affiliations:** Organ Support and Automation Technologies Group, U.S. Army Institute of Surgical Research, JBSA Fort Sam Houston, San Antonio, TX 78234, USA

**Keywords:** controller, closed loop, hemorrhage, resuscitation, machine learning, photoplethysmography, non-invasive, animal study

## Abstract

Hemorrhage is a leading cause of preventable death in military and civilian trauma medicine. Fluid resuscitation is the primary treatment option, which can be difficult to manage when multiple patients are involved. Traditional vital signs needed to drive resuscitation therapy being unavailable without invasive catheter placement is a challenge. To overcome these obstacles, we propose using closed-loop fluid resuscitation controllers managed by non-invasive, intermittent signal sensor inputs to simplify their use in far-forward environments. Using non-invasive, intermittent sensor controllers will allow quicker medical intervention due to negating the need for an arterial catheter to be placed for pressure-guided fluid resuscitation. Two controller designs were evaluated in a swine hemorrhagic shock injury model, with each controller only receiving non-invasive blood pressure (NIBP) measurements simulated from invasive input signals every 60 s. We found that both physiological closed-loop controllers were able to effectively resuscitate subjects out of life-threatening hemorrhagic shock using only intermittent data inputs with a resuscitation effectiveness of at least 95% for each respective controller. We also compared this intermittent signal input to a NIBP cuff and to a deep learning model that predicts blood pressure from a photoplethysmography waveform. Each approach showed evidence of tracking blood pressure, but more effort is needed to refine these non-invasive input approaches. We conclude that resuscitation controllers hold promise to one day be capable of non-invasive sensor input while retaining their effectiveness, expanding their utility for managing patients during mass casualty or battlefield conditions.

## 1. Introduction

Hemorrhage is a critical and urgent medical complication in both civilian and military settings [1], being the leading cause of preventable death from trauma in the pre-hospital setting [2,3,4]. Hemorrhage was attributed partially or entirely to 64% of preventable deaths in the civilian trauma system and 91% in military pre-hospital casualties [5]. Hemorrhagic shock is a life-threatening condition caused by the severe loss of blood volume. When the body loses a significant amount of blood rapidly, the delivery of oxygen to vital organs and tissues is decreased, leading to organ and tissue failure which, if not quickly reversed, results in death [6]. Further difficulty arises due to complex physiological processes that mask typical vital signs associated with hypovolemic shock [7,8], delaying life-saving interventions. Providing effective, efficient and rapid methods of hemorrhage control and interventions will be required for future warfare to improve survival rates.

Current conflicts, such as the Russo-Ukrainian War, have provided an insight on what modern warfare looks like between near-peer adversaries, demonstrating a combat environment where air superiority is contested [9]. In those environments, combat casualties are expected to remain at or near the point-of-injury (POI) for longer durations than seen in previous conflicts. In addition, mass casualties are expected to be prevalent at the POI due to the type of munitions used [10]. These stressors will place additional responsibilities on field clinicians, highlighting the need for technologies that improve casualty outcomes while minimizing the burden on medical personnel in these constrained environments. Autonomous medical systems have the capability to assist by delivering life-saving interventions [11,12]. These interventions are not unique to military settings; they can be applied in scenarios like civilian mass casualties [13], trauma in rural areas where care can be limited [14], or prolonged transportation scenarios resulting from medical care capabilities being inadequate at the location of injury [15].

Due to the type, number and severity of traumatic injuries expected in future peer and near-peer military conflicts, fully autonomous, closed-loop medical systems are being designed to make rapid decisions for singular or multiple casualties, while simultaneously monitoring and continuously managing causalities for improved survivability outcomes. Currently, closed-loop controllers have been successfully developed for fluid resuscitation management [16] as well as for automation of extracorporeal circulatory systems [17]. These closed-loop controllers have shown an improvement in efficiency and stability compared to trained clinicians in providing resuscitation support for hemorrhagic shock to patients needing continuous monitoring of vital signs and fluid administration [18,19]. Our team has previously developed a dual-input fuzzy logic (DFL) controller and validated its ability to resuscitate to a target mean arterial pressure (MAP) in a physical testbed that mimicked different hemorrhage scenarios [20]. Additionally, an adaptive resuscitation controller (ARC) for fluid administration, also developed by our team, has been shown to successfully resuscitate and stabilize swine in a clinically relevant large animal study of hemorrhagic shock [21].

However, these control designs relied on MAP measurements sampled by the controller every 15 s to guide autonomous fluid delivery. This required an arterial catheter to receive the pressure signal in addition to the venous catheter already required for resuscitation, something that can be challenging to achieve in the far-forward environment. If autonomous medical systems can make use of portable non-invasive sensor technologies for vital sign monitoring, these systems will be able to function at the POI through increasing roles of care. Further highlighting their utility, non-invasive sensors have been used to monitor critical vital signs to detect hemorrhagic shock to assist with combat casualty care in an easy to use, portable footprint [22,23]. For the current study, the DFL and ARC resuscitation controllers were modified to process inputs sampled every 60 s, mimicking the constraints imposed by receiving data from an intermittent data stream, such as would be provided by a non-invasive blood pressure (NIBP) cuff. On the other hand, other approaches may be even more advantageous, such as using deep learning (DL) techniques to estimate blood pressure from non-invasive sensor inputs. Techniques for estimating blood pressure from a photoplethysmography (PPG) waveform have been shown to be effective, so similar strategies may be suitable for tracking blood pressure during hypovolemia if features from the PPG waveform can be found to correlate to MAP in these more extreme physiological instances [24,25,26].

The goal of this study was to provide proof-of-concept that a non-invasive input can be used to control autonomous fluid management for resuscitation. A swine model was used to test, validate and provide a basis of comparison between non-invasive ARC and DFL resuscitation controllers. Further, a DL model was developed for the prediction of the MAP utilizing PPG data and was evaluated on the accuracy of its predictions. The results of this study can provide critical information about the utility of autonomous resuscitation controllers when driven by non-invasive inputs that can be used in both clinical and combat casualty situations.

## 2. Materials and Methods

### 2.1. Closed-Loop Controller Design

We tested two controllers with distinct architectures to provide hemorrhagic shock resuscitation in swine. The ARC and DFL controllers were developed and optimized in previous works [16,20,27], so only a brief overview of the key features of each will be covered here, followed by a discussion of an adaptation included for operating with NIBP readings.

The ARC is a novel controller that uses the MAP as the target variable; it was developed using the retrospective analysis of volume responsiveness in swine [28] and further tuned on a hardware-in-loop testbed [29]. For its operation, an initial bolus infusion of 100 mL was given during the first minute of resuscitation to obtain a baseline volume responsiveness measurement. This pressure–volume response was fitted with a linear regression (Δ*P*/Δ*V*), which was then used to predict the infused volume (Δ*V_i_*) required to produce an incremental increase in pressure (Δ*P_i_*). An optimal resuscitation pace (Δ*P*/Δ*T*) of 6 mmHg/min was derived from the iterative tuning conducted previously. Using Δ*P*/Δ*T* and Δ*P*/Δ*V*, an infusion rate (Δ*V*/Δ*T*) was calculated and used for the next infusion step. This process was repeated for each subsequent infusion step, and the linear regression model was updated with each iteration to adapt to the change in the subject’s volume responsiveness. It should be noted that the infusion steps occurred in immediate succession so that the infusion, although supplied at varying rates, was effectively continuous throughout the resuscitation process. Logic rules were imposed to set safety limits and reduce Δ*V*/Δ*T* to 0 once the target MAP was reached. A flow diagram for the ARC is shown in Appendix A.

The DFL controller, which also had the MAP as its target variable, was built using a Sugeno-type fuzzy inference system with the Fuzzy Logic Designer toolbox in MATLAB 2022a (MathWorks, Natick, MA, USA). The two inputs to the controller were *PerformanceError*, defined as the ratio of the current MAP to the target, and *(d/dt)PerformanceError*, which was the average rate of change of the *PerformanceError* over the three most recent data points. The sole output for the DFL was the *InfusionRate,* provided as a decimal value from 0–1, representing the proportion of the maximum infusion rate available; 250 mL/min was the maximum rate in the current work. The *PerformanceError* input was broken into four fuzzy sets, comprised of *VeryLow*, when the pressure was far from the target; *Low*, for intermediate pressure values; *NearSet*, when the pressure was close to the target; and *Set*, when the pressure was at the target. The *(d/dt)PerformanceError* input was divided into 5 fuzzy sets that included *dropFast*, *dropSlow*, *noChange*, *riseSlow* and *riseFast*. The membership functions of all the fuzzy sets for both inputs were plotted and are shown in Appendix A. The output was also categorized into 5 fuzzy sets including *Off*, *MedLow*, *Med*, *MedHigh* and *High*, with *Off* and *High* corresponding to infusion flow rates of 0 and 250 mL/min, respectively. A surface plot of both inputs to the output was generated and is shown in Appendix A.

### 2.2. Non-Invasive Adaptations

The focus of this effort was to evaluate the performance of the controllers when operating with non-invasive inputs. Obtaining blood pressure measurements non-invasively is usually done by applying a blood pressure cuff to an extremity over an arterial vessel. However, NIBP cuffs that could operate reliably with both our test platform and animal models were not available. To work around this problem, we decided to emulate the required NIBP measurements by downsampling the invasive arterial pressure readings that our experimental setup supported. In this approach, instead of sampling the invasive measurements every 15 s as the system was configured to do, we modified it to take a sample every 60 s, which falls within the range of intervals offered by commercially available patient monitors for NIBP readings. This process is diagrammed in Figure 1.

In testing the fluid infusion controllers using a hardware-in-loop testbed (diagram of the setup shown in Appendix A) with the simulated non-invasive inputs, their performance was predictably degraded due to the extended sampling period, exhibiting continuous oscillations around their setpoints. To reduce the impact of the decreased resolution of the controllers’ input signal, we implemented an input preprocessor stage (Figure 1) to condition the non-invasive measurements by generating a set of higher-resolution blood pressure measurements from the non-invasive readings. This preprocessor would apply linear extrapolation to the 2 most recently acquired non-invasive measurements to estimate 4 pressure measurements projected over the next 60 s—i.e., these estimated measurements had a sampling period of 15 s. Each of the estimated measurements would then be provided by the preprocessor to the controllers at the appropriate rate (every 15 s). The preprocessor would correct its estimation with every new measurement. The end result of this process was an input sampling rate artificially increased from the slower NIBP rate to one that matched the interval at which the controllers were designed to operate (i.e., 15 s).

Finally, while testing the DFL controller with the preprocessor stage, we noticed that it was more affected by the initial estimated measurements generated by the preprocessor than the ARC design. Upon starting to operate, the preprocessor did not have enough pressure readings to generate an upsampled set of measurements, so it only passed through a raw NIBP reading. Due to its design, the ARC incorporated an initial 1 min-long “buffering period” to acquire sufficient data to update its internal prediction models. As it happened, by the time that the buffering was completed, the upsampling preprocessor stage had also started generating estimated pressure measurements; therefore, the ARC did not rely much on the initial raw NIBP pressure. On the other hand, the DFL controller did not incorporate any such buffering period in its design, so it fully relied on the initial NIBP measurement immediately upon starting to operate. To improve the DFL controller operation with the preprocessor stage, we further modified it to start all its resuscitations by infusing a 100 mL bolus over 1 min, which would give time for the preprocessor to start operating. After that initial bolus, the standard DFL control logic would take over and continue the resuscitation.

### 2.3. Animal Model

This research was conducted in compliance with the Animal Welfare Act, implementing Animal Welfare regulations and the principles of “The Guide for the Care and Use for Laboratory Animals” [30]. The Institutional Animal Care and Use Committee (IACUC) at the United States Army Institute of Surgical Research approved all the research conducted in this protocol, A-24-003. The facility where this research was conducted is fully accredited by AAALAC International.

#### 2.3.1. Overview of Animal Preparation and Instrumentation

This was a prospective, open-label, non-randomized, pre-clinical pilot study to demonstrate proof-of-concept for multiple physiological closed-loop controllers for hemorrhagic shock resuscitation, utilizing a modification of a previously used swine (*sus scrofa domestica*) model [21]. There was no control group, as the goal of the study was to demonstrate that the controllers could effectively resuscitate the subject. We used *n* = 10 intact female Yorkshire crossbred swine that were around four months old and weighed approximately 40 kg. Swine were chosen as the large animal model as it is a widely accepted animal model for hemorrhage studies due to the swine’s physiological similarities to humans, including the cardiovascular system [31,32]. The sample size was decided arbitrarily based on resource constraints, and all the animals in the study protocol were included in the final data analysis. All the animals were maintained at a surgical plane of anesthesia throughout the study, first by the continuous inhalation of isoflurane (0–5%), followed by total intravenous anesthesia (TIVA) using ketamine (0–10 mg/kg/h) and midazolam (0–2 mg/kg/h); buprenorphine SR (0.24 mg/kg) was given for analgesia. All the anesthetics were actively titrated to effect throughout the protocol, and all the animals remained in the supine position for the duration of the study. The primary outcome measures for the study included the controller performance metrics described below.

Initially, each animal was tranquilized with intramuscular tiletamine-zolazepam (4–6 mg/kg), anesthetized with isoflurane as above, mechanically ventilated to maintain an end-tidal carbon dioxide of 40 ± 5 mmHg, and a foley catheter was placed for continuous urinary output monitoring. A blood oxygen saturation sensor was placed into the mouth after the cheeks were shaved to improve the PPG signal clarity. Five electrocardiography leads were placed on the lateral surfaces of the limbs and the chest. A percutaneous sheath introducer (Arrow International, Morrisville, NC, USA) was placed into the right jugular for the placement of a Swan–Ganz pulmonary artery catheter (Edwards Lifesciences, Irvine, CA, USA), and a triple-lumen central venous catheter (Arrow International, Morrisville, NC, USA) was placed into the left jugular for TIVA administration. An arterial catheter (Arrow International, Morrisville, NC, USA) was placed into either the left or right carotid to measure the arterial blood pressure (ABP), while another arterial line was placed into either the left or the right femoral artery for blood sampling. Additionally, a femoral venous line was introduced for both hemorrhage and resuscitation. A neonatal NIBP cuff was placed on the animal’s tail to non-invasively capture the blood pressure data (MAP_Cuff_) for comparative analysis. Following the line placement, there was a minimum 10 min stabilization period. A laparotomy was then performed for a surgical splenectomy utilizing surgical clips (Weck^®^ Hem-o-lok^®^, Teleflex Medical, Wayne, PA, USA) to ligate splenic vessels for removal. After closing the abdomen in two layers with a suture and staples, all the animals underwent a 30 min stabilization period. During this 30 min stabilization period, the anesthetic delivery was changed from isoflurane gas to TIVA using Ketamine (0–10 mg/kg/h) and Midazolam (0–2 mg/kg/h). This 30 min stabilization period was then followed by a 10 min data-gathering hold period.

#### 2.3.2. Hemorrhage Event

Following the stabilization, the animals were hemorrhaged in a stepwise manner using a peristaltic pump controlled by an automated hemorrhage decision-table software written in MATLAB 2022a (“AutoBleed”), which allowed for the precise control of the bleed. AutoBleed withdrew blood until a targeted MAP was reached. The system continued to both remove and reinfuse blood as needed to maintain the targeted MAP for each hemorrhage pressure ‘step’. To prevent coagulation, AutoBleed controlled an additional pump which mixed a citrate, phosphate, dextrose and adenine (CPDA-1, Terumo BCT, Lakewood CO) solution with the withdrawn whole blood (WB) at a 1:7 ratio. The target arterial pressures for the hemorrhage ‘steps’ were 75, 65, 55, 45 and 35 mmHg, with the first hemorrhage target based off the baseline MAP post-stabilization hold—e.g., if the baseline MAP was 65 mmHg, then the first hemorrhage target was 55 mmHg. Each stepdown was succeeded by a 10 min hold and an arterial lactate draw. A lactate analysis was performed every 10 min, utilizing an iSTAT CG4^+^ cartridge (Abbott Point of Care, Princeton, NJ, USA). Once AutoBleed reached the bottom ‘step’, it maintained the targeted MAP at 35 mmHg until either 90 min lapsed or an arterial lactate threshold ≥ 4 mmol/L was reached.

#### 2.3.3. Controller-Based Resuscitation

AutoBleed was stopped after either the lactate threshold was reached, or (if the lactate threshold was reached before the 35 mmHg target pressure was reached) after a 10 min hold at 35 mmHg. A full set of labs were then drawn, and a calcium chloride bolus (1 g/10 mL) was administered to offset the calcium-chelating capacity of the CPDA-1 anticoagulant that was about to be re-infused. Two resuscitation controllers, ARC or DFL, were tested during the fluid resuscitation, with each controller utilizing *n* = 5 swine. The first DFL-based resuscitation did not utilize the initial 100 mL bolus in its design and is identified separately throughout the results. The animals were not randomized, but instead controller assignments were done in an alternating fashion for this pilot study. The fluid resuscitation controlled by either the ARC or DFL utilized the autologous blood previously hemorrhaged, with the oldest blood given back first. A maximum of 1 L of WB was given to any animal, after which the infusate was switched to lactated Ringer’s (LR) solution. If less than 1 L of WB was collected during the hemorrhage, then only the amount that was drawn would be given back before switching to LR. Both controllers were set to reach the pre-hemorrhage baseline MAP or a target of 65 mmHg, whichever was lower, and maintain it throughout the 60 min hold period. Our previous study showed that maintaining a damage control resuscitation target MAP of 65 mmHg worked and was within the Joint Trauma System guideline recommendations [21,33].

#### 2.3.4. Controller and Data Collection Setup

A diagram of how data were sent to the controller during the animal study is shown in Figure 1. Prior to the experiments starting, the systems operator was informed which resuscitation controller was being used. The operator oversaw initiating either the AutoBleed, ARC, or DFL at the time points stated in the protocol. During the resuscitation phase of the study, the operator also switched the controller to LR when the system notified that the pre-set volume of WB had been re-infused. Over the course of the study, a patient monitor captured the traditional vital signs, waveforms and NIBP readings. On all the animals, a HemoSphere monitoring device was attached to the Swan–Ganz catheter to record the cardiac output and mixed venous oxygen saturation. A purpose-built data acquisition system was utilized to capture the data. A sampling rate of 500 Hz was utilized for analog waveforms, while 1/5 Hz was utilized for all other digital data recordings. This system also monitored and recorded the flow rates of the peristaltic pumps used, along with the weight scale readings for the hemorrhage volumes.

### 2.4. Controller Performance Metrics

Arterial pressure data for each animal were analyzed separately, and the average MAP was calculated across all subjects. Commonly used controller metrics were selected to objectively evaluate both the ARC and DFL’s performance at reaching and sustaining the target pressure [34,35,36]. These metrics previously described [16] include the effectiveness, resuscitation effectiveness, rise-time efficiency, median performance error (MDPE), wobble, target overshoot, area above and below the target MAP, mean and median infusion rates, and infusion rate variability. The resuscitation effectiveness was previously introduced [21], which was a modification of the existing effectiveness metric. The effectiveness is defined as the portion of resuscitation time where the arterial pressure stayed within 5 mmHg below and above the target MAP. Unlike the effectiveness, the resuscitation effectiveness does not penalize for instances where the target was exceeded, since this is less concerning during hemorrhagic shock resuscitation. The raw data were processed and performance metrics calculated using Jupyter Notebook and Python, while GraphPad Prism 10.3 (La Jolla, CA, USA) was used to analyze and visualize the final data.

### 2.5. PPG Analysis

#### 2.5.1. Data Processing

For the development of a DL model for the non-invasive prediction of the MAP using PPG, swine hemorrhage data collected from our previous animal studies [21] were used in addition to the data collected in this study. Analog and digital data were recorded at 500 Hz and 1/5 Hz, respectively. The analog data were downsampled to a frequency of 100 Hz for the development of the DL models. The PPG signal was filtered using a 2nd order Butterworth bandpass filter with cutoff frequencies of 0.5 Hz and 10 Hz, which has been shown to be effective for the removal of spurious artifacts from PPG signals [37]. After filtering, the outliers in the PPG signal were identified by calculating the interquartile range (IQR). PPG signals that fell outside 1.5 times the IQR in the positive and negative directions were replaced by linear interpolation. After filtering the outliers, the 1st through 4th derivatives were calculated from the PPG signal, based on previous work where higher order derivatives aided with predicting blood pressure values using PPG data from a patient database [38]. To preprocess the PPG time series data into an input for the DL model, they were segmented in intervals of 20 s for both training and testing the DL model. Ultimately, the DL model input consisted of 5 features (PPG waveform and its aforementioned derivatives) and 2000 samples.

#### 2.5.2. Deep Learning Model Development

The DL model developed for the prediction of the MAP from a non-invasive PPG waveform was a hybrid convolutional neural network (CNN) and a long short-term memory (LSTM) network. The DL model consisted of five inputs, the filtered PPG waveform and its four derivatives, and a single output, MAP (Figure 2). Due to the limited availability of data, the “leave one subject out” (LOSO) cross-validation technique was used to train, validate and test the models. The swine data gathered in this study were split into three different categories: the training set, the validation set and the testing set. The training set was exclusively used to train, and the validation set was used to tune the hyperparameters of the DL model. The training set was 2/3 of the entire data collected in this study, while the validation and testing each used equal parts of the remaining 1/3 of the data. The training data were supplemented with physiological PPG waveform data of 12 additional swine from a previous hemorrhage and resuscitation study [21]. The supplemental data underwent the same data processing methodology.

#### 2.5.3. Deep Learning and NIBP Performance Metrics

Four metrics were used to compare the ground-truth MAP recorded by the patient monitor with the PPG-to-MAP DL model-predicted MAP (MAP_PPG_) and MAP_Cuff_ during the baseline, hemorrhage, hypovolemic hold, and resuscitation phases of the experiment. The first metric, the coefficient of determination (R^2^), was calculated to demonstrate the proportion of variance between the two MAP values. Secondly, the root mean squared error (RMSE) was calculated to estimate how well the DL model predicted the actual value. Thirdly, an accuracy score was calculated to see how well the MAP_PPG_ and MAP_Cuff_ fell within the acceptable range of the MAP (±10 mmHg), as determined by international standards for blood pressure measuring devices [39]. Finally, the reliability of the MAP values were calculated based on the percentage of time where a non-zero signal was obtained for the MAP_Cuff_ or MAP_PPG_.

### 2.6. Correlative Analysis

To compare the signal quality between the MAP and other potential signal inputs for resuscitation controllers, we correlated data trends for each signal input’s ability to predict the MAP. First, correlations were assessed as the ground-truth MAP was streamed to the controller computer (MAP_Streamed_). A second set of correlations was assessed when the ground-truth MAP signal was downsampled to one datapoint every 60 s and the same value was held for the intermittent readings (MAP_Intermittent_). A final set of correlations was assessed for downsampling the ground-truth MAP to one datapoint every 60 s and then supplementing the missing data via the extrapolation method described in Section 2.2 (MAP_Upsampled_). In addition, we evaluated correlations between the ground-truth, continuous MAP, MAP_PPG_ and MAP_Cuff_. For each, linear trends were calculated to get a goodness of fit and 95% prediction intervals. Bland–Altman plots were generated with 95% confidence intervals to assess the prediction similarity to the ground-truth MAP. All the analyses were performed using GraphPad Prism 10.3 (GraphPad, La Jolla, CA, USA).

## 3. Results

### 3.1. Comparison of ARC and DFL Controllers in Live Animal Study

There was variance observed in the baseline MAPs of the 10 animals used in the study. One had a starting MAP of 85 mmHg, eight animals had MAPs between 60–65 mmHg, and a single animal had the lowest pressure at 55 mmHg. There was no obvious explanation for these discrepancies other than the unique physiology of the individual animals. During the hemorrhage phase of the study, eight animals reached a lactate level of 4 mmol/L or higher, either prior to or upon reaching the shock target MAP of 35 mmHg; the remaining two animals did not reach a lactate level of 4 mmol/L until after entering the 90 min hypovolemic hold: one 30 min in and the other at 60 min. Due to the baseline MAPs, most animals had a target MAP for resuscitation of 60–65 mmHg, except the one with the lowest baseline, which had the target MAP set to 55 mmHg instead. During this phase, nine animals were infused with 800 mL to 1 L of autologous blood, while one only received 518 mL; this subject was the same one with the highest baseline MAP described above. The administered volume of LR solution varied for each animal over the 60 min resuscitation hold. However, one animal did not receive the LR solution as enough WB was available for the resuscitation window. Despite the subject variability, the controller results were minimally affected, highlighting their robustness and adaptive capabilities.

The performance of the ARC and DFL controllers (*n* = five swine each) can be assessed from the overall data trends as well as the controller performance metrics measured during the fluid resuscitation. Both controllers were successful at reaching the target MAP and maintaining at or above the target MAP for the entire 60 min resuscitation window (Figure 3). Overall, the effectiveness was 83 ± 12% for ARC compared to 91 ± 5.5% for DFL (Figure 4A). However, this metric penalized the controller performance for overshooting the target, which is less of a concern during hemorrhage resuscitation (as the clinical goal of the resuscitation would be for the subject’s MAP to continue rising unassisted). Therefore, we also evaluated a resuscitation effectiveness metric that only subtracts from the performance at pressures below the target MAP. With this metric, the ARC resuscitation effectiveness was 95 ± 1.7% compared to 96 ± 2.0% for DFL (Figure 4B). Another data trend difference between the ARC and DFL was found in the rate at which the target MAP was initially approached during the resuscitation, with DFL more aggressively recovering the MAP. This was quantified by the rise-time efficiency metric, with the ARC requiring 2.2 ± 0.58 min compared to a quicker 1.4 ± 0.41 min for DFL (Figure 4B). The metrics for areas above and below the target MAP were quantified for both controllers to evaluate the cumulative magnitude of controller over- and undershoot during the 60 min resuscitation window. The ARC had a higher area above the target MAP metric than DFL (124 ± 70 mmHg × min vs. 62 ± 60 mmHg × min), but a smaller area below the target MAP (69.9 ± 18 mmHg × min vs. 85 ± 48 mmHg × min) (Figure 4C). Another difference between the ARC and DFL was related to the controller noise across the 60 min window, with DFL more closely maintaining the target MAP. We quantified this by the MDPE and wobble, with DFL having lower values for each (−0.26 ± 2.1% MDPE; 1.6 ± 0.06% wobble) compared to the ARC (2.3 ± 1.9% MDPE; 2.2 ± 0.58% wobble) (Figure 4D). While DFL’s stability trended higher, its overall infusion rates were similar to the ARC. The median infusion and mean infusion rates for the ARC were 9 ± 17 mL/min and 32 ± 18 mL/min, respectively, compared to 10 ± 9 mL/min and 37 ± 13 mL/min, respectively, for DFL (Figure 4E). Lastly, the infusion rate variability was 26 ± 10 mL/min for the ARC and similarly at 30 ± 7.2 mL/min for DFL (Figure 4E). These performance metrics as well as others are summarized in Appendix A.

### 3.2. Comparison Between Non-Invasive and Invasive ARC Performance

We previously evaluated the ARC with a continuous invasive arterial input signal during hemorrhage resuscitation in swine [21]. As such, we can compare the effects of the intermittent, non-invasive signal feed to the ARC (present study) relative to a continuous invasive signal input (previous study) on the overall controller performance. It is worth noting that these were two separate animal protocols with slight experimental differences; the previous study (i) resuscitated to the target MAP with WB for a full 10 min followed by LR instead of limiting to two units of WB during resuscitation; (ii) due to nationwide ketamine shortages, more than half of the previous protocol animals were anesthetized with other agents (i.e., isoflurane or propofol); and (iii) the previous protocol resuscitation phase continued for 2 h. For ease of comparison, we only focused on the first 60 min of the continuous ARC resuscitation in the previous study. Overall, the controller reached the target MAP in both the invasive and non-invasive input configurations with similar rise-time efficiency metrics (Figure 5C). However, there was a slightly higher noise level evident with the invasive controller design (Figure 5) as measured by the MDPE and wobble (Figure 5D). The area above and below the target MAP scores were comparable for both controller configurations, further highlighting how similarly the controller performed with both types of inputs (Figure 5E).

### 3.3. Comparison of Intermittent Data Streams

#### 3.3.1. Comparison of Continuous MAP to Simulated Non-Invasive MAP

To further evaluate the effect of intermittent data and other possible data streams on the closed-loop resuscitation performance, we compared the continuous MAP signal to the intermittent MAP, as well as to other non-invasive inputs, such as the MAP_Cuff_ and MAP_PPG_ as described in Section 2.6. The MAP_Streamed_ introduced some signal noise, but the goodness of fit score was still 0.847, and 95% confidence intervals were −4.4 to 4.2 mmHg compared to the continuous MAP (Figure 6A,D). The MAP_Intermittent_ showed a reduced goodness of fit of 0.803, and 95% confidence intervals widened to −5.3 to 5.4 mmHg compared to the continuous MAP (Figure 6B,E). For the MAP_Upsampled_, the goodness of fit compared to the continuous MAP was actually further reduced to 0.672, and the width of 95% confidence intervals was further widened to −7.7 to 7.2 mmHg, indicating more noise was introduced to the signal feed by this process (Figure 6C,F). This was not the intended outcome for this extrapolation signal upsampling methodology, but it highlights the robustness of the closed-loop controller designs to maintain a high performance with increased signal noise.

#### 3.3.2. Comparison of Non-Invasive Blood Pressure Cuff MAP to Continuous MAP

The swine were instrumented with an arterial catheter to gather the MAP and had an NIBP cuff placed to gather MAP_Cuff_ data for a total of nine swine; the tenth animal’s tail was too short for a cuff. The MAP_Cuff_ was compared to the gold-standard, continuous MAP signal (Table 1). The R^2^ of the MAP_Cuff_ vs. continuous MAP measurements was, on average, 0.445. The range for R^2^ was between 0.010 and 0.733. The RMSE was 26 mmHg on average, ranging between 17 mmHg and 44 mmHg. In addition, the average accuracy of the MAP_Cuff_ measurements, based on the percentage of time pressure was within 10 mmHg of the gold standard, was 54.5%, and ranged between 37.9% and 76.1%. Finally, the reliability of the MAP_Cuff_ measurements were calculated based on the percentage of time where a non-zero signal was read by the NIBP, and was on average, 71.7% of the time, ranging between 55.6% and 84.3%.

The MAP_Cuff_ measurements were compared against the gold-standard pressure from the arterial catheter. The results for the swine subjects with a strong and poor correlation between the two measurements are shown in Figure 7, highlighting their differences. In addition, the upper and lower boundaries of the acceptable range for the MAP_Cuff_ measurements relative to the continuous MAP are also shown. The poor correlation example had various regions where the NIBP was unable to read the blood pressure, resulting in a much lower reliability score at 66.0%. Poor NIBP readings across all the subjects were more often at low pressures. The MAP_Cuff_ measurements on this subject were found to be within the acceptable accuracy range from the continuous MAP reading only 41.3% of the time. The strong correlation example only became zero at the hypovolemic pressure phase of the study, resulting in a higher reliability score (78.7%). The MAP_Cuff_ readings on this animal were within an acceptable range of the MAP readings 73.0% of the time. All other swine had various ranges of reliability and accuracies, presented in Table 1, and similar time plots for the remaining swine are shown in Appendix A.

#### 3.3.3. Comparison of MAP_PPG_ to Continuous MAP

Performance metrics for MAP_PPG_ predictions compared to the continuous MAP are summarized in Table 2. The MAP_PPG_ predictions and continuous MAP for both test subjects are shown in Figure 8, along with the upper and lower boundaries of the acceptable range from the continuous MAP. The poor correlating test result had various regions where the signal predicted pressure values outside the tolerable range, resulting in a lower accuracy (23.7%). The strong correlating example trended close to the continuous MAP value throughout the study, only falling outside of the acceptable range a few times, resulting in a higher accuracy (90.6%). Both the low- and high-performance MAP_PPG_ predictions had a reliability of 100% as PPG was present throughout the study and the DL model was able to make a prediction for every segment of time input into the model.

## 4. Discussion

Hemorrhage control and resuscitation are the highest impact medical therapies for improving mortality in potentially survivable trauma cases, for both civilian patients and military casualties. During fluid resuscitation, WB and/or other fluids are infused to increase the circulating volume, raise the MAP levels, restore the cardiac output and improve the end-organ perfusion. Automating this procedure with closed-loop controlled systems extends the settings where effective fluid resuscitation can be offered by lowering the skill threshold for providers and reducing the cognitive burden required when monitoring one or more patients for long periods of time [40,41,42]. However, remote, pre-hospital settings often lack the invasive high-resolution arterial pressure signals typically available in emergency departments and operating rooms and are instead usually limited to only a non-invasive blood pressure cuff. In this study, we evaluated the performance of an ARC and DFL, two automated control architectures, at providing fluid resuscitation in a swine hemorrhage model when limited to an intermittent, upsampled MAP input as well as limited infusate resources, which may be more representative of these settings. We also investigated various methods for obtaining a NIBP reading in our swine model and compared them to gold-standard invasive arterial line data.

Both the ARC and DFL were able to effectively resuscitate animal subjects to the target MAP levels when inputs were constrained to a single reading once every 60 s, and upsampled to provide a predicted input every 15 s. The controllers succeeded in this, despite being limited to up to 1 L of WB and relying on LR afterward. As the MAP is generally less responsive to LR compared to WB, this reflects positively on the controllers’ capability to achieve their goals even when resources are limited. Accordingly, these controllers could be anticipated to improve the delivery of resuscitative medical care in any potentially resource-limited circumstance, such as in mass casualty incidents, wilderness medicine and rural medicine.

When not penalized for overshooting the target, both controllers achieved mean resuscitation effectiveness scores above 90%, demonstrating their ability to maintain the animals at or above their target pressure for a majority of the time. In most of the metrics examined, both controllers performed similarly. On average, the ARC required slightly lower infusion rates while DFL demonstrated a slightly improved rise-time efficiency and generally held the MAP more stable (as indicated by the improved wobble and divergence scores). The ARC had a higher average area above the target metric (about twice that of DFL), while DFL had a higher average area below the target. The differences in the rise-time efficiency, general stability, and area metrics seem to indicate that, while DFL reached the target quicker and better avoided overshooting the target, the ARC was better suited for responding to intermittent drops in MAP and was more responsive in restoring the MAP to the target, even if this often resulted in an overshoot. In all cases, the controllers were able to successfully resuscitate the animal subjects out of life-threatening hemorrhagic shock, despite rather significant biological variability; because the real-world-use cases for these controllers all include the potential for large inter-individual variability, we believe this capability bodes well for the ultimate safety profile of these devices.

Comparing the ARC performance from the current study to previous results that utilized a high-fidelity invasive pressure input demonstrated little, if any, notable differences. When considering the increased noise of the difference plot comparing the intermittent, upsampled input provided to the controller to the invasive arterial data, this demonstrates a stout robustness in performing consistently, in some respects, independent from the input source. Although not conclusive, it seems to be the case that, so long as the input pressures are supplied at a consistent rate, the ARC will provide resuscitation in a way that can compensate for some degree of inaccuracy in the actual measurement. This may help to expand the threshold of acceptable accuracies for non-invasive, predictive measures moving forward. As such, we evaluated various alternative non-invasive inputs that could be used by the ARC in future experiments. The MAP_Upsampled_ signal was lower in signal noise compared to the potential MAP_Cuff_ or MAP_PPG_ signals. Given both controllers’ strong performance with the current intermittent input, we think it is likely that the controller performance will remain strong even if relying on some of these even higher variable sensor inputs. However, these alternative inputs can be improved for future use. In the case of the NIBP cuff, the placement in the swine was challenging given their truncated extremity anatomy, requiring the placement of the cuff around the animals’ tail. Other animal models, such as a non-human primate model, may be used as an alternative due to blood pressure cuffs being able to be placed at similar anatomical locations to humans [43,44,45]. The placement on a limb in human subjects may result in a higher signal accuracy [46], but issues with use at hypotensive states may remain, which will continue to present a challenge for hemorrhagic shock resuscitation applications [47,48,49]. On the other hand, the preliminary work presented on predicting the MAP from PPG presents a solution for hypovolemic conditions where blood pressure cuffs tend to fail, as seen in this study. In addition, PPG can be gathered continuously, resulting in estimated MAP measurements generated at higher sampling rates. We acknowledge that the MAP_PPG_ is in the early stages of development, as there is a distinct lack of hemorrhage and resuscitation datasets available to further develop the models. Additional swine hemorrhage and resuscitation data will continue to be captured at the USAISR to improve these DL models.

While the conducted research study highlights the utility of closed-loop controllers for fluid resuscitation with intermittent sensor input, there were some limitations with the study design. First, the animal study was a pilot in nature, resulting in few animals being used (i.e., only five subjects for each controller). The subject variability is high in live animal studies, making it harder to identify more nuanced differences in performance of the two controller designs currently. Animal studies with larger sample sizes in the future will be able to better identify statistically significant strengths and weaknesses for each controller design. Also, as previously stated, the small subject quantity impacted the MAP_PPG_ predictions as well. More data will be needed to improve these models in the future and reduce the variability currently experienced across subjects. To aid with this, we are retrospectively evaluating swine study datasets for building more robust models. A second limitation of this study is that the comparison between the non-invasive and invasive ARC variants has caveats, as these experiments were conducted with slightly different animal models. We truncated previous data to a one-hour duration for a more even comparison, but different amounts of WB given may have impacted some of the performance metrics presented. Finally, another limitation of this study is that the intermittent signals used were derived from invasive sensor measurements, which are less applicable to pre-hospital situations. This was done due to known challenges with NIBP cuff placement in swine and MAP_PPG_ predictions being undeveloped prior to this animal study. NIBP cuff placement would be more feasible in a non-human primate hemorrhage study, which could be pursued in future research efforts. This study still presents evidence that these controllers can function with intermittent data feeds, but future work will be needed through additional animal studies to pair these controllers with non-invasive sensor inputs. 

## 5. Conclusions

Next to the rapid control of active bleeding, timely fluid resuscitation remains the most effective treatment for hemorrhagic shock. Closed-loop controllers such as those presented in this research effort can aid with off-loading the cognitive burden and lowering the skill threshold for administering life-saving treatment in the far-forward military and/or pre-hospital environment, shortening both the time and distance between the point-of-injury and effective intervention. We found that both the ARC and DFL controller configurations were capable of reaching a target MAP and maintaining at this set point for at least one hour. This was accomplished with an imposed infusate limitation of up to two units of WB supplemented by LR solution, and controllers receiving an intermittent, upsampled blood pressure signal rather than direct high-fidelity data from an invasive arterial line. This was done to explore the potential for future controller designs to function with fully non-invasive sensor signals such as NIBP cuffs or PPG waveforms. Work is currently planned to continue improving the MAP_PPG_ predictions by utilizing additional animal data to improve training. The next stage of the resuscitation controllers is also under development to incorporate vasopressor administration, and testing in a large animal study is expected in the measurable future. This capability will be added to address volume non-responsive patients or when resource constraints indiscriminately limit the availability of fluids.

While in the near-term, the development of these controllers is focused on the future battlefield and related resource-constrained settings, it seems feasible that with further development, this technology could be used to deliver safe and effective care for hemorrhagic shock in otherwise well-resourced settings like hospitals. Accordingly, the development of these technologies has broader implications for the future of healthcare delivery, including potential improvements in both the safety and the cost of care, with the ultimate goal of improving the quality of medical care for trauma-related hemorrhage. 

## 6. Patents

Eric J. Snider and Saul J. Vega are co-inventors on a patent filed on the adaptive resuscitation controller technology (ISR 23-45-US02: 18/628,287).

## Figures and Tables

**Figure 1 bioengineering-11-01296-f001:**
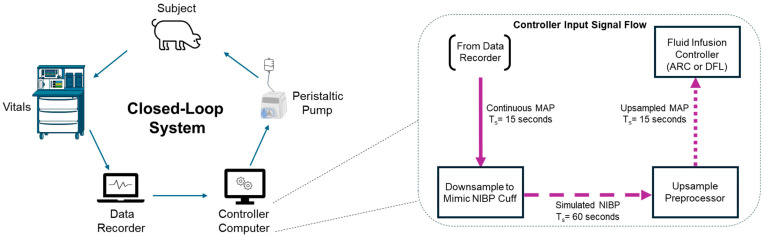
Diagram of closed-loop controller setup and data streaming during animal procedures. The animal subject’s vital signs were captured by a patient monitor and sent to a data recorder system. Then, data were passed to the controller computer wherein the data were downsampled and then upsampled (Sampling Time = T_S_). Decisions from the fluid infusion controller (ARC or DFL) were sent to a peristaltic pump which administered fluid to the subject.

**Figure 2 bioengineering-11-01296-f002:**
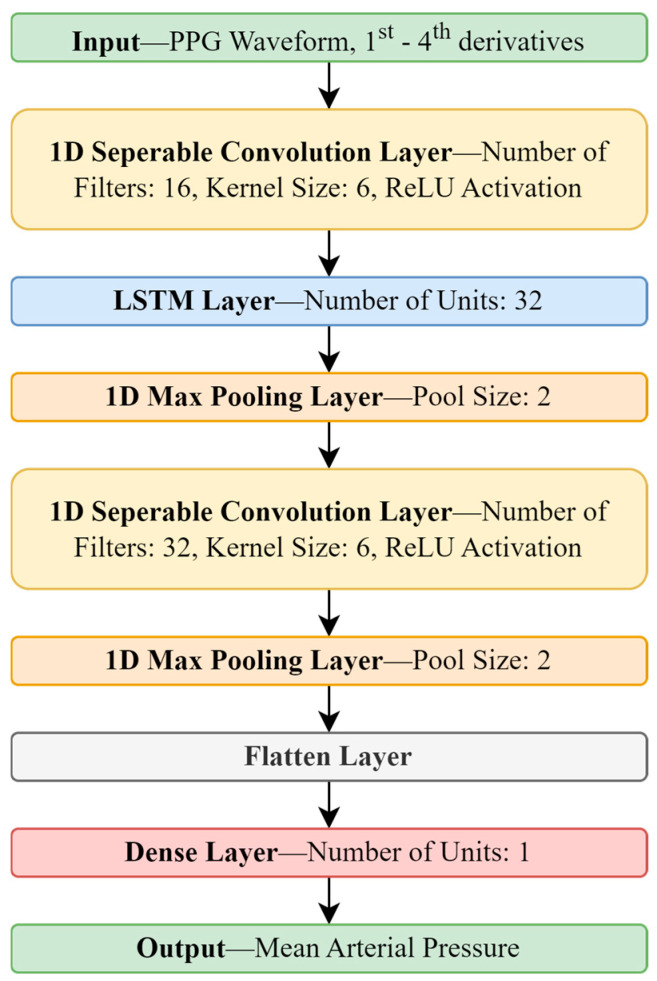
Deep learning (DL) model architecture for the prediction of the mean arterial pressure (MAP) from photoplethysmography (PPG).

**Figure 3 bioengineering-11-01296-f003:**
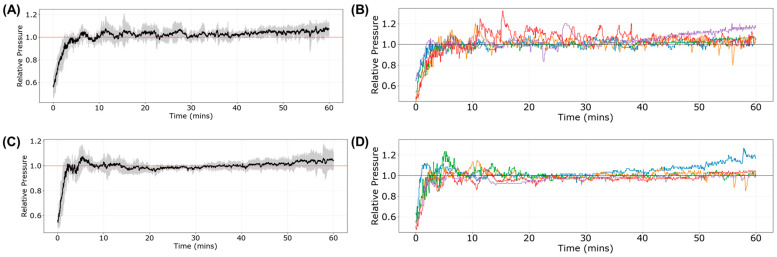
Overview of (**A,B**) the adaptive resuscitation controller (ARC) and (**C,D**) dual-input fuzzy logic (DFL) resuscitation with simulated non-invasive, intermittent signal input. (**A**,**C**) average results with shaded regions denoting standard deviation, and (**B**,**D**) pressure vs. time results for each individual subject (*n* = five swine each), each identified by a different color. Pressure values are relative to their target MAP for each subject. For DFL, the subject identified by the blue line did not receive an initial 100 mL bolus of whole blood (as mentioned in Section 2.2).

**Figure 4 bioengineering-11-01296-f004:**
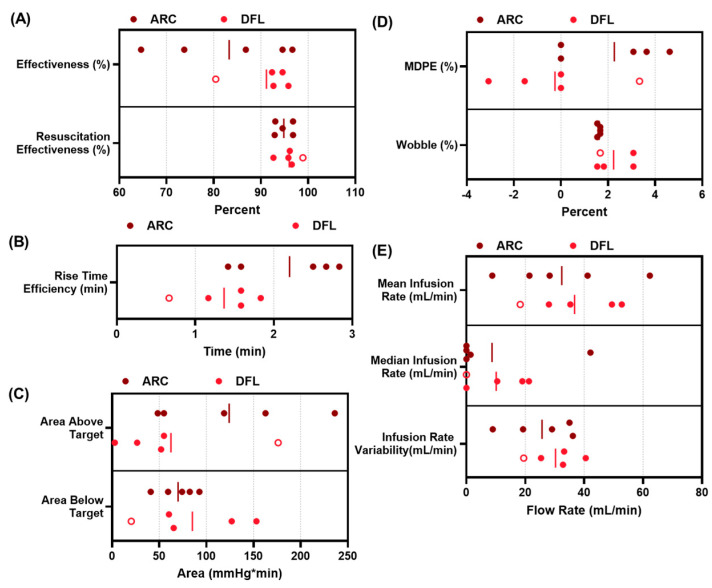
Summary of the performance metrics for the ARC and DFL. (**A**) Effectiveness and resuscitation effectiveness, (**B**) rise-time efficiency, (**C**) area above and below target, (**D**) median performance error (MDPE) and wobble, and (**E**) mean infusion rate, median infusion rate, and infusion rate variability. Lines denote averages and individual subjects are shown (*n* = 5 for each). For DFL, one subject did not receive an initial 100 mL bolus of whole blood and is shown as an open circle on each sub-figure.

**Figure 5 bioengineering-11-01296-f005:**
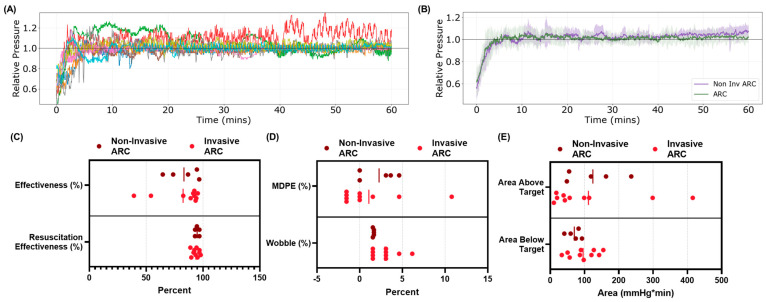
Summary of performance metrics for invasive and non-invasive ARC. (**A**) Relative pressure vs. time results for invasive ARC-based resuscitation for each individual subject (*n* = 10); each subject is identified by a different line color. (**B**) Average results for invasive (*n* = 10) and non-invasive (*n* = 5) ARC-based resuscitation. Shaded region denotes standard deviation. Performance metric comparison between invasive and non-invasive ARC for (**C**) effectiveness and resuscitation effectiveness, (**D**) MDPE and wobble, and (**E**) area above and below target MAP. Lines denote averages and individual subjects are shown (*n* = 5 non-invasive; *n* = 10 invasive).

**Figure 6 bioengineering-11-01296-f006:**
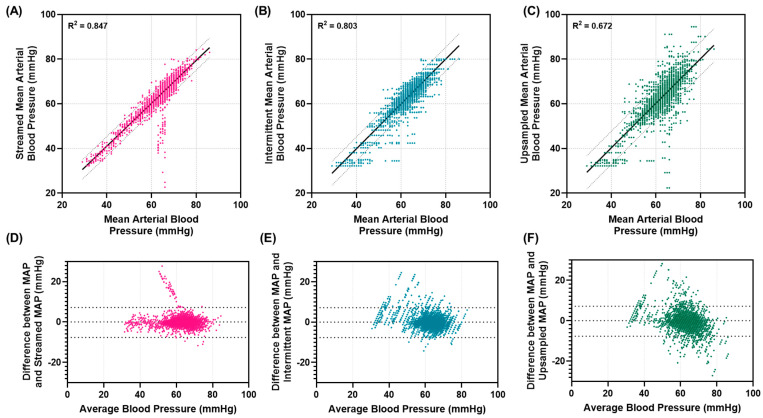
Effects of intermittent sampling on correlation to continuous MAP signal. Results are shown comparing the mean arterial pressure to (**A**,**D**) the pressure after being streamed to the controller computer (MAP_Streamed_), (**B**,**E**) after downsampling the controller signal to one value received every 60 s (MAP_Intermittent_), and (**C**,**F**) the signal having undergone upsampling (MAP_Upsampled_). (**A**–**C**) Correlation plots are shown for each with the goodness of fit coefficient for linear regression shown as well as 95% confidence intervals. (**D**–**F**) Bland–Altman plots for differences between each signal vs. the average signal, with 95% confidence intervals shown for each.

**Figure 7 bioengineering-11-01296-f007:**
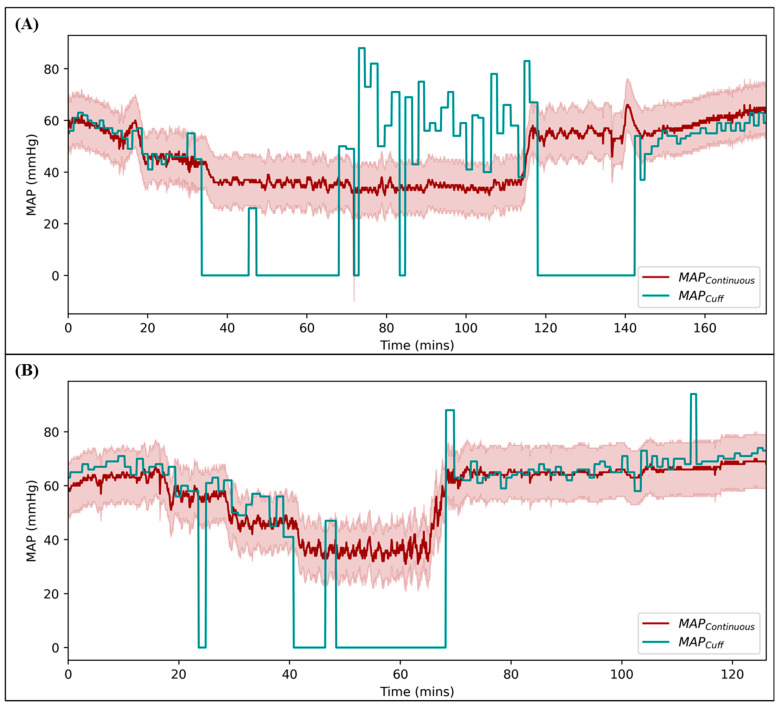
MAP_Cuff_ vs. continuous MAP for a representative (**A**) poor and (**B**) strong correlation swine subject result. The shaded region represents the acceptable accuracy range for the continuous MAP compared to the MAP_Cuff_ in each example.

**Figure 8 bioengineering-11-01296-f008:**
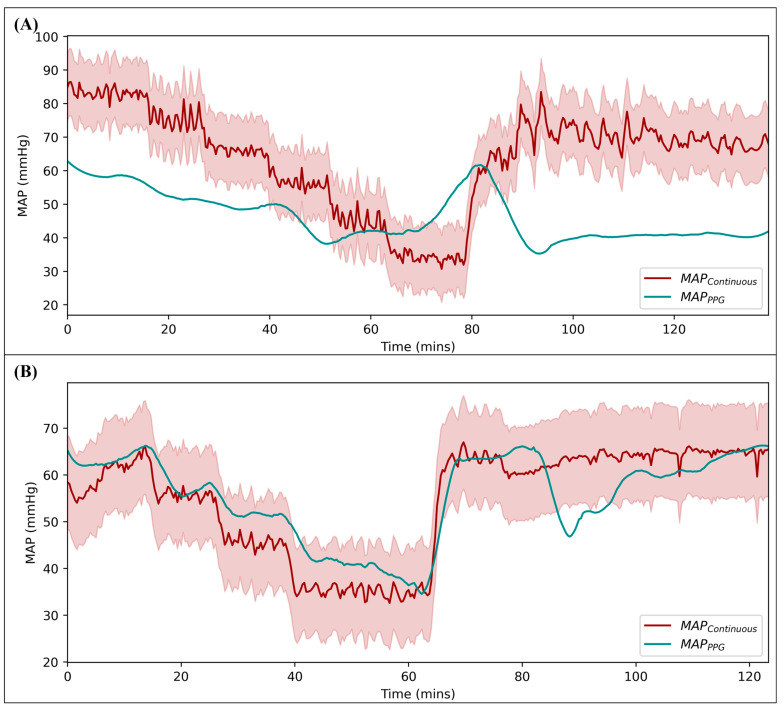
Predicted MAP_PPG_ compared to continuous MAP for a (**A**) poor correlation and a (**B**) strong correlation swine test subject. The shaded region represents the acceptable accuracy range for the continuous MAP compared to the MAP_PPG_ in each example.

**Table 1 bioengineering-11-01296-t001:** MAP_Cuff_ Performance Metrics of Swine Subjects Compared to Continuous MAP.

	R^2^	RMSE (mmHg)	Accuracy (%)	Reliability (%)
Swine 1	0.733	23.2	47.1	75.1
Swine 2	0.508	17.4	76.1	84.2
Swine 3	0.0516	44.1	28.6	55.5
Swine 4	0.755	19.0	73.0	78.6
Swine 5	0.474	23.1	57.5	76.0
Swine 6	0.760	19.1	72.1	76.9
Swine 7	0.00966	30.3	41.2	66.0
Swine 8	0.266	34.3	37.9	58.9
Swine 9	0.448	22.8	57.0	73.3
Average	0.445	25.9	54.5	71.7

**Table 2 bioengineering-11-01296-t002:** MAP_PPG_ Performance Metrics of Two Test Subjects Compared to Continuous MAP.

	R^2^	RMSE (mmHg)	Accuracy (%)	Reliability (%)
Subject 1	0.053	22.8	23.7	100
Subject 2	0.738	5.9	90.6	100

## Data Availability

The datasets presented in this article are not readily available because they have been collected and maintained in a government-controlled database that is located at the US Army Institute of Surgical Research. As such, these data can be made available through the development of a Cooperative Research & Development Agreement (CRADA) with the corresponding author. Requests to access the datasets should be directed to Eric Snider, eric.j.snider3.civ@health.mil.

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
