# Peer review of "In Vivo Evaluation of Two Hemorrhagic Shock Resuscitation Controllers with Non-Invasive, Intermittent Sensors"

_bioengineering, 2024, doi:10.3390/bioengineering11121296_

Round 1

Reviewer 1 Report

Comments and Suggestions for Authors

In this work, authors proposed an In vivo evaluation of two hemorrhagic shock resuscitation controllers with non-invasive intermittent sensors. Closed loop controllers can aid with offloading the cog. burden. Further, two controller designs were evaluated.  The approach showed evidence of blood pressure tracking. Overall the work is good, but i have some comments:

(1) In the introduction section, authors mentioned about control designs relied on continuous arterial waveform signals. Can you elaborate this ?

(2) In section-2, 2.1, authors mentioned about hardware in loop testbed. Provide a picture related to this statement.

(3) If the authors have used some hardware or testing apparatuses, then these should be given in the paper as figures to help readers easily understand. Most of the things are just in language text.

(4) Fig. 1 resolution is not good. 

(5) Fig. 2 (a)(c) and Fig. 4(b) have visible background of another set of data. What is this shaded region ? Please discuss here in this comment.

(6) Have the authors only used Matlab as software or have they used nay other software for simulations ?

Author Response

In this work, authors proposed an In vivo evaluation of two hemorrhagic shock resuscitation controllers with non-invasive intermittent sensors. Closed loop controllers can aid with offloading the cog. burden. Further, two controller designs were evaluated.  The approach showed evidence of blood pressure tracking. Overall the work is good, but i have some comments:

(1) In the introduction section, authors mentioned about control designs relied on continuous arterial waveform signals. Can you elaborate this ?

We appreciate the reviewer taking the time to review our manuscript. As suggested, more details have been added about the signal and how it is used by the controller to guide resuscitation in the introduction section.

(2) In section-2, 2.1, authors mentioned about hardware in loop testbed. Provide a picture related to this statement.

This was published in prior studies by our team where the testbed was used to downselect and tune the controllers evaluated in a large animal model in this paper. As such, we decided to add a figure diagramming the test platform as a supplementary figure per suggestion to avoid distracting from the animal testing results which are the focus of this effort.

(3) If the authors have used some hardware or testing apparatuses, then these should be given in the paper as figures to help readers easily understand. Most of the things are just in language text.

We have added a diagram that explains the testing setup for the closed loop controller used during live animal studies as Figure 1 in the paper. We hope this helps the reader better understand the process.

(4) Fig. 1 resolution is not good. 

The figure has been reimported as a PNG with a dpi of 400 to address the resolution issue.

(5) Fig. 2 (a)(c) and Fig. 4(b) have visible background of another set of data. What is this shaded region ? Please discuss here in this comment.             

Thank you for pointing this out, the shaded region is the standard deviation of the signals. This is stated in the figure captions of the associated figures mentioned.

(6) Have the authors only used Matlab as software or have they used any other software for simulations ?

Prior to this live study, during the simulation phases of the project, we only used custom MATLAB code integrated with a physical testbed to develop and tuned the fluid resuscitation controllers. No other simulation software was used. This physical testbed is mentioned in the Introduction and the Materials and Methods sections (with additional references), but a diagram of this setup will now be included as a supplemental figure in the manuscript.

Reviewer 2 Report

Comments and Suggestions for Authors

Abstract:

  1. Line 11-12: The introduction succinctly frames hemorrhage as a critical issue. A clearer emphasis on the novelty of the "non-invasive, intermittent-sensor controllers" could make the abstract more compelling.
  2. Line 19-20: The finding that both controllers effectively resuscitated subjects is significant. However, stating more specific quantitative results could enhance the impact.
  3. Line 23-25: The conclusion about potential battlefield utility is well-written but could briefly mention the primary limitations or areas for further research.

Introduction:

  1. Line 31-33: Excellent framing of the problem. Referencing specific statistics about hemorrhage-related deaths in military or civilian scenarios could strengthen this section.
  2. Line 36-39: The mention of physiological masking delaying interventions is important. Consider citing specific studies or mechanisms for clarity.
  3. Line 41-44: Modern warfare context is relevant but tangential. It might be better positioned in a separate paragraph about applications.
  4. Line 49-50: Autonomous systems are emphasized well. Expanding briefly on their broader implications (e.g., in civilian trauma) could be valuable.

Materials and Methods:

  1. Line 95-97: Clear summary of the controllers’ key features. A diagram illustrating the controller mechanisms would enhance understanding.
  2. Line 129-135: The adaptation of non-invasive measurements is innovative. Clarifying why the specific 60-second interval was chosen would improve the rationale.
  3. Line 175-183: The animal model setup is comprehensive. However, justifying the use of Yorkshire swine (e.g., anatomical similarities to humans) would strengthen the methodology.

Results:

  1. Line 339-342: Noting physiological variability among subjects is important. However, further elaboration on how this impacted the study’s findings would add depth.
  2. Line 363-364: The resuscitation effectiveness of 95–96% is impressive. A brief comparison to conventional methods would contextualize this achievement.
  3. Line 374-375: Quantifying noise metrics is insightful. Discussing their potential clinical implications (e.g., patient safety) would make this finding more impactful.

Discussion:

  1. Line 503-504: The relevance of fluid resuscitation is well-articulated. A stronger emphasis on how this research fills gaps in current practices would be beneficial.
  2. Line 523-525: Highlighting the controllers’ performance under limited resources is commendable. Speculating on real-world implications (e.g., rural healthcare) could enhance this section.
  3. Line 556-557: The discussion of challenges with NIBP cuffs in swine is thorough. Suggesting alternative validation models could add value.

Limitations:

  1. Line 565-567: Acknowledging the small sample size is crucial. Providing a specific plan for scaling up (e.g., multicenter trials) would demonstrate foresight.
  2. Line 576-577: The limitation of using intermittent signals derived from invasive measurements is critical. Proposing specific improvements for future studies strengthens credibility.

Conclusion:

  1. Line 587-589: The conclusion effectively summarizes key findings. Highlighting the broader societal impact (e.g., cost-effectiveness) could enhance its resonance.
  2. Line 602-604: The mention of future work with vasopressors is forward-thinking. Suggesting a timeline or roadmap for development would provide clarity.

General Comments

  • The research is impactful and well-structured but could benefit from more visual aids (e.g., flowcharts or comparative tables).
  • Ethical compliance is thoroughly addressed, which adds credibility.
  • Expanding the discussion on clinical implications and providing a clearer roadmap for translational research would enhance the manuscript's relevance.

Author Response

Abstract:

  1. Line 11-12: The introduction succinctly frames hemorrhage as a critical issue. A clearer emphasis on the novelty of the "non-invasive, intermittent-sensor controllers" could make the abstract more compelling.

We appreciate the reviewer so carefully evaluating our manuscript and providing constructive feedback. We agree with the reviewer and have added a statement to provide clearer emphasis on the novelty in the abstract.

  1. Line 19-20: The finding that both controllers effectively resuscitated subjects is significant. However, stating more specific quantitative results could enhance the impact.

Thank you for the feedback, the quantitative results have been added to the abstract.

  1. Line 23-25: The conclusion about potential battlefield utility is well-written but could briefly mention the primary limitations or areas for further research.

Thank you for the feedback, we have added the primary limitations of these controllers regarding overall resuscitation strategies in the abstract.

Introduction:

  1. Line 31-33: Excellent framing of the problem. Referencing specific statistics about hemorrhage-related deaths in military or civilian scenarios could strengthen this section.

We have added statistics for both civilian and military hemorrhage death percentages for potentially survivable deaths.

  1. Line 36-39: The mention of physiological masking delaying interventions is important. Consider citing specific studies or mechanisms for clarity.

Thank you for the suggestion, we have added specific study references related to the compensation/masking mechanisms mentioned.

  1. Line 41-44: Modern warfare context is relevant but tangential. It might be better positioned in a separate paragraph about applications.

As recommended, information regarding modern warfare is in a separate paragraph with information regarding situations where possible applications would be beneficial.

  1. Line 49-50: Autonomous systems are emphasized well. Expanding briefly on their broader implications (e.g., in civilian trauma) could be valuable.

Thank you for the feedback, we added information regarding civilian trauma that could cause prolonged care.  Instances such as mass casualties, rural areas and wilderness traumas where these lifesaving interventions would be beneficial.

Materials and Methods:

  1. Line 95-97: Clear summary of the controllers’ key features. A diagram illustrating the controller mechanisms would enhance understanding.

We have added an additional supplementary figure highlighting the logic of the ARC controller’s design and a supplementary figure highlighting the membership functions for the DFL controller. We hope that helps with understanding the controller mechanisms.

  1. Line 129-135: The adaptation of non-invasive measurements is innovative. Clarifying why the specific 60-second interval was chosen would improve the rationale.

This was selected to mimic the sampling rate of a non-invasive blood pressure cuff. We state this in the methods section to justify this decision.

  1. Line 175-183: The animal model setup is comprehensive. However, justifying the use of Yorkshire swine (e.g., anatomical similarities to humans) would strengthen the methodology.

We have added a statement clarifying this decision as well as associated references to further strengthen the methodology.

Results:

  1. Line 339-342: Noting physiological variability among subjects is important. However, further elaboration on how this impacted the study’s findings would add depth.

The controller designs handle subject variability well so that the subject variability did not impact results of the study much. We added a statement in this section alluding to the results not being affected by this variability.

  1. Line 363-364: The resuscitation effectiveness of 95–96% is impressive. A brief comparison to conventional methods would contextualize this achievement.

This is a difficult comparison to make. Some studies have looked at conventional resuscitation for comparison to closed loop approaches, but animal experiment details, severity of hemorrhage, fluid delivery strategies, etc… differ. As such, making that comparison is not accurate to the data in this study. The wide range of performance metrics will also make this comparison challenging to interpret given these differences. While outside the scope of this study, this I something we will consider doing in future research efforts.  

  1. Line 374-375: Quantifying noise metrics is insightful. Discussing their potential clinical implications (e.g., patient safety) would make this finding more impactful.

Thank you for this suggestion. We have added a section to the discussion highlighting the likely strong safety profile of these controllers in view of their ability to be effective despite significant biological variability.

Discussion:

  1. Line 503-504: The relevance of fluid resuscitation is well-articulated. A stronger emphasis on how this research fills gaps in current practices would be beneficial.

Thank you for the feedback, we have added additional emphasis on how this would be beneficial by including a few additional references on the impact of this type of work.

  1. Line 523-525: Highlighting the controllers’ performance under limited resources is commendable. Speculating on real-world implications (e.g., rural healthcare) could enhance this section.

Thank you for this input, we have added a sentence highlighting the potential for these controllers to improve the delivery of care in resource-limited settings.

  1. Line 556-557: The discussion of challenges with NIBP cuffs in swine is thorough. Suggesting alternative validation models could add value.

Thank you for the feedback, we have added a statement about additional models that will allow anatomical placement similar to humans for validation as well as references to said animal models.

Limitations:

  1. Line 565-567: Acknowledging the small sample size is crucial. Providing a specific plan for scaling up (e.g., multicenter trials) would demonstrate foresight.

We added scaling up animal studies as a next step for this in the limitations sections to address this request by the reviewer.

  1. Line 576-577: The limitation of using intermittent signals derived from invasive measurements is critical. Proposing specific improvements for future studies strengthens credibility.

Great suggestion. We believe a non-human primate model would be the best path forward to resolve the non-invasive blood pressure cuff limitations in swine. We have added reference to this in the limitations section.

Conclusion:

  1. Line 587-589: The conclusion effectively summarizes key findings. Highlighting the broader societal impact (e.g., cost-effectiveness) could enhance its resonance.

Thank you for this comment – we have added a paragraph to the conclusion highlighting the potential impact of this technology on the quality of trauma care.

  1. Line 602-604: The mention of future work with vasopressors is forward-thinking. Suggesting a timeline or roadmap for development would provide clarity.

Thank you for the suggestion, we have added more details addressing the development and future testing of the vasopressor additions.

Round 2

Reviewer 1 Report

Comments and Suggestions for Authors

Authors have answered all comments satisfactorily.